# Synergistic antibacterial effect of hydroxyl radicals generated by the combination of hypochlorous acid and UV irradiation

Hwa Yong Lee[1,2], Han Bit Lee[1,2], Younghee Kim[1]*

1 Department of Convergence Engineering, Graduate School of Venture, Hoseo University, Seoul, Republic of Korea, 2 Research & Development Department, Enputech Co., Ltd., Gwangju-si, Gyeonggi-do, Republic of Korea

* yhkim514@hoseo.edu

## Abstract

Livestock farms are at risk of exposure to various environmental pollutants, particularly airborne viruses that can cause infectious diseases. Hydroxyl radicals (•OH) are well-known for their strong bactericidal and virucidal properties and are widely applied in disinfection processes. However, their efficacy is significantly diminished in the presence of organic substances. This study investigated the bactericidal effects of hydroxyl radicals generated from hypochlorous acid (HOCl) under UV irradiation and evaluated their resistance to quenching by airborne organic matter. Rose Bengal (RNO) dye was used as a probe to detect •OH radical generation, while yeast extract served as a representative organic contaminant. RNO bleaching efficiency increased in a concentration-dependent manner under UV irradiation, confirming the formation of hydroxyl radicals. However, in the presence of yeast extract, this bleaching effect was drastically reduced, indicating that organic compounds can interfere with radical activity. The bactericidal effects of UV light and HOCl were independently evaluated using Salmonella as a model organism. The presence of organic matter significantly reduced the bactericidal efficacy of both UV and HOCl treatments when applied separately. In contrast, combined exposure to HOCl and UV irradiation demonstrated a 10% increase in bacterial reduction and halved the required exposure time, regardless of HOCl concentration. These findings highlight the synergistic bactericidal potential of HOCl and UV irradiation and support their applicability in airborne bacterial disinfection under realistic environmental conditions.

## 1. Introduction

The importance of air quality in livestock farms has grown alongside increasing emphasis on animal welfare. Poor air quality can lead to respiratory diseases, stress, and weakened immune function, particularly in densely populated breeding environments. Airborne particles carrying viruses can be easily transported by wind, leading

**Data availability statement:** All relevant data are within the manuscript.

**Funding:** This study is financially supported by Korea Institute of Planning and Evaluation for Technology in Food, Agriculture and Forestry(Funding No. 821034-3).

**Competing interests:** The authors have declared that no competing interests exist.

to epidemics among livestock [1]. Outbreaks of infectious diseases result in substantial economic losses due to mass mortality and may also pose health risks to farm workers. Therefore, maintaining clean air in livestock facilities is strongly recommended [2,3].

Sprayed disinfectants based on electrolyzed water may serve as effective tools for removing airborne microbes in various settings, including the food industry, healthcare, residential environments, and agriculture. These solutions are advantageous because they do not cause facility corrosion, chemical toxicity, or environmental pollution [4,5].

Electrolyzed water can be classified as acidic, slightly acidic, or alkaline. Among these, slightly acidic electrolyzed water (SAEW), which primarily contains hypochlorous acid (HOCl), is known for its high disinfection efficiency and broad-spectrum antimicrobial activity [6]. Slightly acidic hypochlorous acid can be generated through the electrolysis of a dilute solution of hydrochloric acid (HCl) and sodium chloride (NaCl), which supplies the necessary chloride ions. Among various chlorine-based disinfectants (HOCl, ClO$^-$, Cl$_2$), HOCl produced at a pH range of 5.0 to 6.5 is more bactericidal than hypochlorite ions (OCl$^-$) [7,8]. However, recent studies have reported that at concentrations exceeding 300 ppm and with prolonged exposure, HOCl may exert cytotoxic and genotoxic effects on human cells [9].

Hypochlorous acid exhibits broad-spectrum antimicrobial activity, effectively targeting both gram-positive and gram-negative bacteria, as well as fungi and viruses, including avian influenza (AI), foot-and-mouth disease (FMD), and African swine fever (ASF) [10–15].

Ultraviolet (UV) irradiation represents another environmentally friendly method for sterilizing airborne microorganisms. Among the UV spectrum, UVC radiation is particularly effective, as it is strongly absorbed by biomolecules such as DNA and proteins, thereby exhibiting potent disinfecting effects [16,17].

When hypochlorous acid is exposed to UV light, it undergoes a chain reaction that generates hydroxyl radicals (•OH) and reactive chlorine species (RCSs), such as chlorine radicals (Cl•) and hypochlorite radicals (OCl•) [18–20]. The formation of chlorinated byproducts during this process can have both beneficial and adverse effects: while OCl$^-$ and Cl$^-$ contribute to disinfection, they may also lead to the production of disinfection byproducts (DBPs) during water treatment [21,22].

Hydroxyl radicals are highly potent oxidizing agents and belong to the class of reactive oxygen species (ROS). Despite their short lifespans, they are capable of decomposing organic matter and exhibit strong bactericidal activity [23]. However, under realistic farm conditions, the presence of organic contaminants can significantly hinder the disinfection efficiency of •OH radicals [24,25]. While the synergistic disinfection effects of chlorinated compounds such as sodium hypochlorite and chlorine under UV irradiation have been previously studied, similar investigations involving hypochlorous acid (HOCl) remain limited [26–28].

In this study, the bactericidal efficacy of HOCl, a chlorine-based compound, was examined under UV irradiation. Additionally, the study evaluated its practical applicability in livestock farming environments, with a particular focus on the inhibitory effects of airborne organic matter on the activity of hydroxyl radicals.

## 2. Materials and methods

### 2.1 Preparation of test materials

Hypochlorous acid was generated by electrolytic water generator (model: Pu:lox HAS-1563, Enputech Co., Ltd. Republic of Korea). 1% diluted hydrochloric acid using 100 ml of 37% hydrochloric acid solution (Cas no.:7647-01-0, Deoksan Chemical, Republic of Korea) was mixed with 2% NaCl solution in a 1:1 ratio, and then diluted with 4 times the amount of water and injected into the generator. The 5% hypochlorous acid solution was generated at 5V/ 20A current condition and used to prepare 80, 120, and 150 ppm, respectively. Concentrations of hypochlorous acid were measured using free chlorine and total chlorine ultra-high range meter (Model: HI 97771, Hanna Instruments, Woonsocket, RI, USA). The generated hypochlorous solution is slightly acidic with a pH ranging from 5.5 to 6.5 (measured using a pH meter Model: ST-3100-F, Ohaus, Parsippany, NJ, USA).

As the organic materials, 25 wt% yeast extract solution was prepared by adding 25 g of yeast extract (KisanBio, Republic of Korea) to 75 g of sterile distilled water. *Salmonella enterica subsp. enterica* was obtained from the Korean Collection for Type Cultures (KCTC) with strain number KCTC 2421, originally sub-deposited from the American Type Culture Collection (ATCC) with strain number ATCC 19585. It is incubated at 35°C for 24 hours in Tryptic soy broth (TSB) culture medium. The cultured bacterial population was measured after incubation at 35°C for 24 hours with plating 0.1 mL onto tryptic soy agar plates.

### 2.2 OH radical generation

OH radicals was generated by 80, 120, 150 ppm of hypochlorous acids and UV light. 24W UV-C germicidal lamp (short-wave 254nm UV lamp, ozone-free, Philips, measured value 0.12 μW/cm$^2$ at a distance of 1 meter) was used.

### 2.3 RNO decomposition test

p-Nitrosodimethylaniline(RNO, CAS no: 138-89-6, Sigma-Aldrich) was used to investigate radical effect of hypochlorous acid and UV [29]. 20 mg of RNO was thoroughly dissolved in 1 L of distilled water for 1.5 hours. UV lamp was switched on and then RNO absorbance was observed for 300s using a UV-VIS spectrophotometer (UV/Vis/NIR Spectrophotometer model: V-770, Jasco International Co. Ltd, Tokyo, Japan). For RNO bleaching 80, 120, and 150 ppm of hypochlorous acid solutions were used and 2 mL of 25% yeast extract was added to determine the inhibitory effect of organic matter.

### 2.4 Antibacterial effect of hypochlorous and UV

Transparent chamber with an internal size of 1m × 1m × 1m was equipped with two 0.2 mm spray nozzles (18 ml/min per unit), a 24W UV-C germicidal lamp and a wireless fan as shown in Fig 1. Airborne bacteria were diffused using a nebulizer (NE-C28, Omron Healthcare Co., Ltd., Kyoto, Japan) and the air sampler (Spin Air samplers Revodix) was used for collecting samples on 90 mm petri dish. 2 ml (about 9 × 106 CFU/mL) of *Salmonella* solution diluted 1:100 in TSB was injected into the nebulizer and sprayed in the chamber. UV was irradiated for 1 minute and 80, 120, 150 ppm of hypochlorous acid test was used and reaction time was 1 minute. To evaluate synergistic effect, hypochlorous acid and UV light were exposed simultaneously. In addition and 2 mL of 25% yeast extract was added to determine the inhibition of organic matter on antibacterial effect.

## 3. Results and discussion

### 3.1 RNO bleaching by OH radicals

RNO exhibits strong absorbance at 440 nm, appearing bright yellow in color. As the –NO group in RNO is oxidized by hydroxyl radicals (•OH), its color gradually fades; thus, it is widely used as an indicator for •OH radical generation. In experiments conducted to confirm whether hypochlorous acid produces •OH radicals under UV irradiation (Fig 2), no

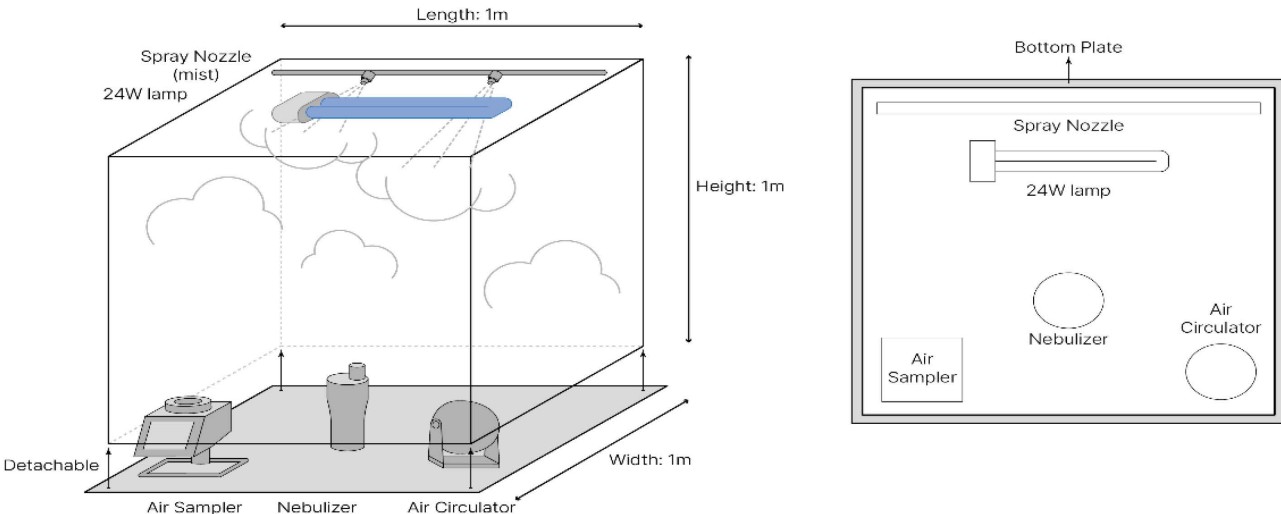

**Fig 1. Experimental apparatus.**

change in absorbance at 440 nm was observed with UV exposure alone, indicating that •OH radicals were not generated. However, upon the addition of hypochlorous acid, the primary absorption peak shifted from 440 nm to 400 nm.

The initial absorbance of RNO varied depending on the concentration of hypochlorous acid, which may be attributed to radical formation by hypochlorous acid itself. As UV exposure time increased, significant changes in absorbance were observed: at 80 ppm, most of the RNO was oxidized after 180 seconds; at 120 ppm, a marked decrease occurred after 120 seconds; and at 150 ppm, the effect was evident after only 60 seconds. These results suggest that radical formation was positively correlated with the concentration of hypochlorous acid. Although previous studies have reported that RNO may exhibit absorbance at other wavelengths due to the formation of oxidized derivatives [29], such shifts were not observed in this study.

Proteins and peptides are known to be degraded into amino acids by hydroxyl radicals (•OH) [30]. In this study, yeast extract—rich in proteins and amino acids—was used to evaluate the effect of organic matter on the oxidative activity of •OH radicals generated by hypochlorous acid under UV irradiation. In the RNO decomposition experiment with added yeast extract, the generation of •OH radicals was confirmed at both 80 ppm and 150 ppm HOCl concentrations. However, at 80 ppm, the oxidative efficiency of •OH radicals was significantly diminished, and complete RNO degradation was not observed even after 300 seconds of reaction time compared to the control without yeast extract (Fig 3). After 180 seconds, a new absorbance peak emerged at 300 nm and continued to increase over time. This phenomenon was more pronounced at 150 ppm, and since it was not observed in the absence of yeast extract, it is presumed to result from the oxidation of the organic components in the yeast extract.

### 3.2 Bactericidal effect of OH radical induced by UV or HOCl

Under organic matter-free conditions, hypochlorous acid (HOCl) demonstrated bactericidal efficacy at all tested concentrations—80, 120, and 150 ppm—resulting in bacterial reductions of $4.24 \pm 0.14$, $4.56 \pm 0.12$, and $4.95 \pm 0.07$ $\log_{10}$ CFU/m³, respectively (Table 1).

HOCl exhibits the strongest bactericidal activity among chlorine-based disinfectants [31], primarily due to its small molecular size and neutral charge, which facilitate cellular penetration more effectively than other chlorine species [4,32]. Its antimicrobial activity against *Salmonella* is largely attributed to its ability to disrupt essential cellular proteins, thereby

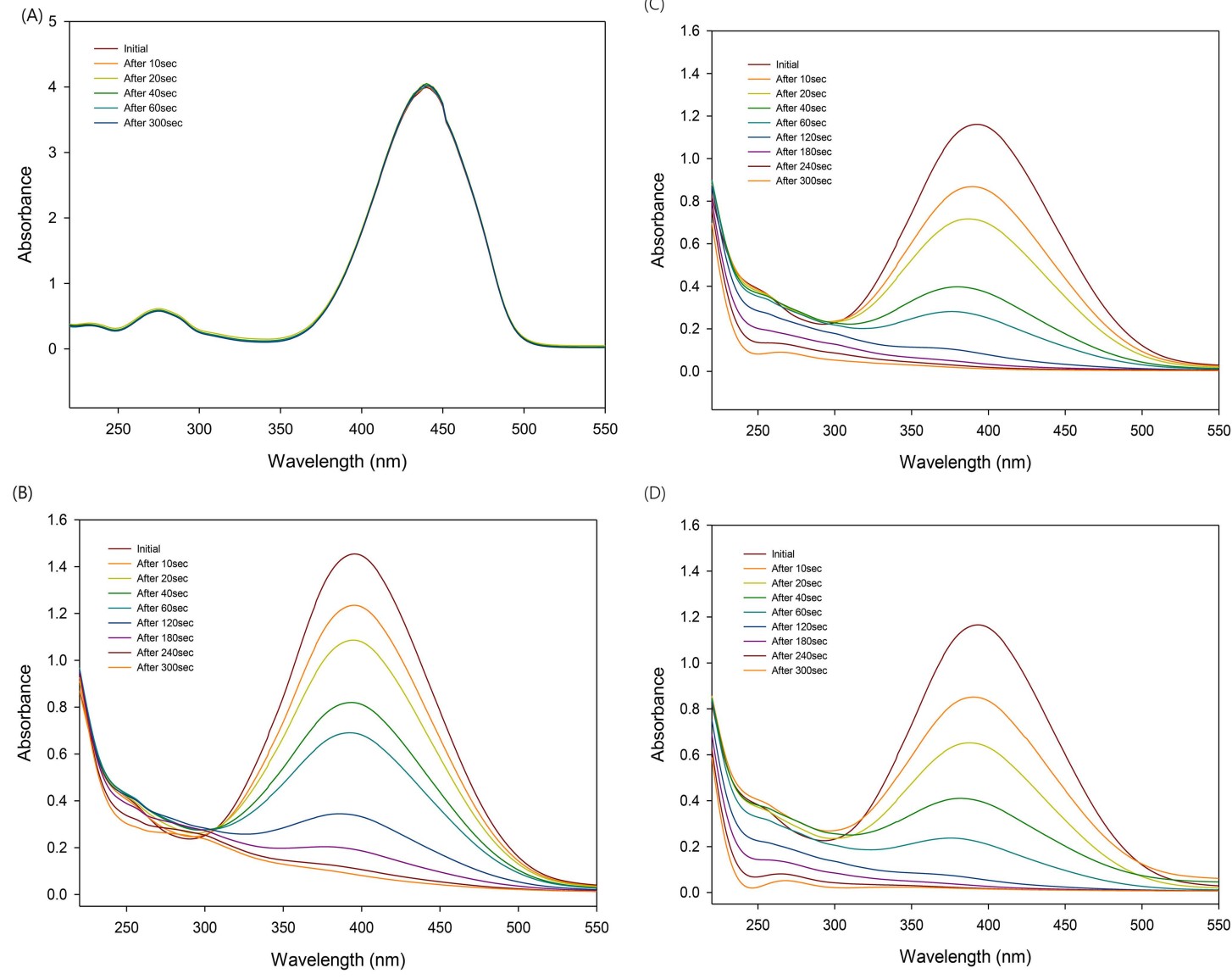

**Fig 2. Bleaching of RNO with UV irradiation in hypochlorous acid solution (A) H₂O (B) 80 ppm (C) 120 ppm and (D) 150 ppm.**

inhibiting bacterial growth and division. This is due to its high reactivity with proteins involved in transcription and translation processes, ultimately leading to a significant reduction in total cellular protein content [33].

Ultraviolet (UV) radiation—particularly UVC—is also widely recognized for its bactericidal properties. UVC light is effectively absorbed by biomolecules such as DNA, RNA, and proteins, leading to microbial inactivation through disruption of nucleic acid replication [34,35]. As shown in Table 1, UV exposure alone resulted in a bacterial reduction of $4.06 \pm 0.15$ $\log_{10}$ CFU/m³, a level comparable to the effect of 80 ppm HOCl.

However, in the presence of organic matter, the bactericidal efficacy declined significantly in both treatment conditions. The reduction in cell count decreased by 21.9% for UV alone and by up to 35% for HOCl alone. Specifically, at 120 and 150 ppm HOCl, bacterial reductions decreased from $4.56 \pm 0.12$ and $4.95 \pm 0.07$ $\log_{10}$ CFU/m³ to $3.64 \pm 0.07$ and $3.99 \pm 0.05$

(A)

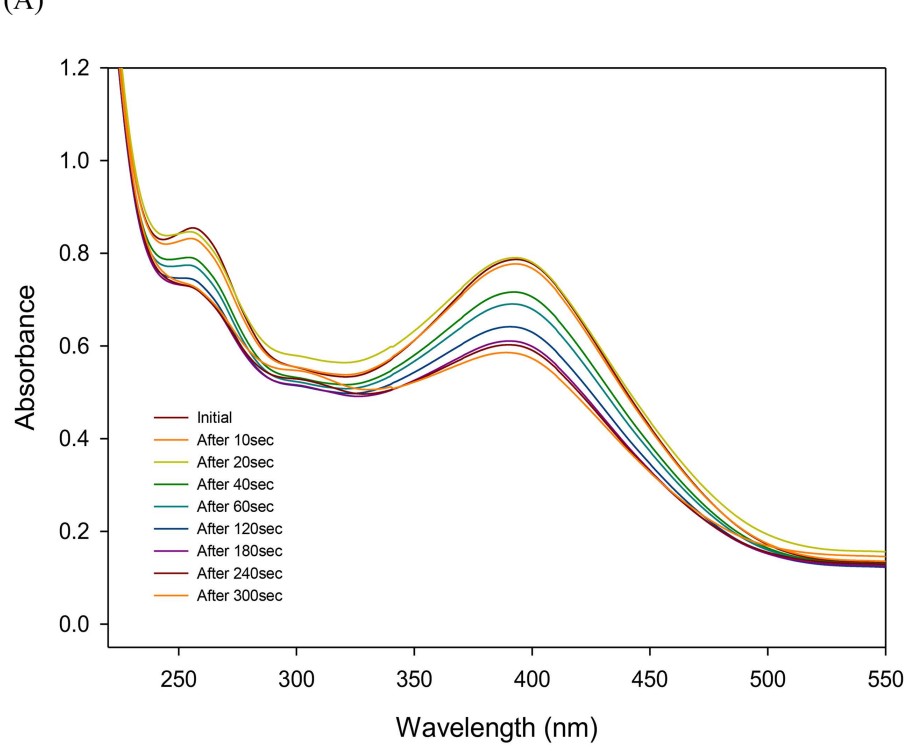

(B)

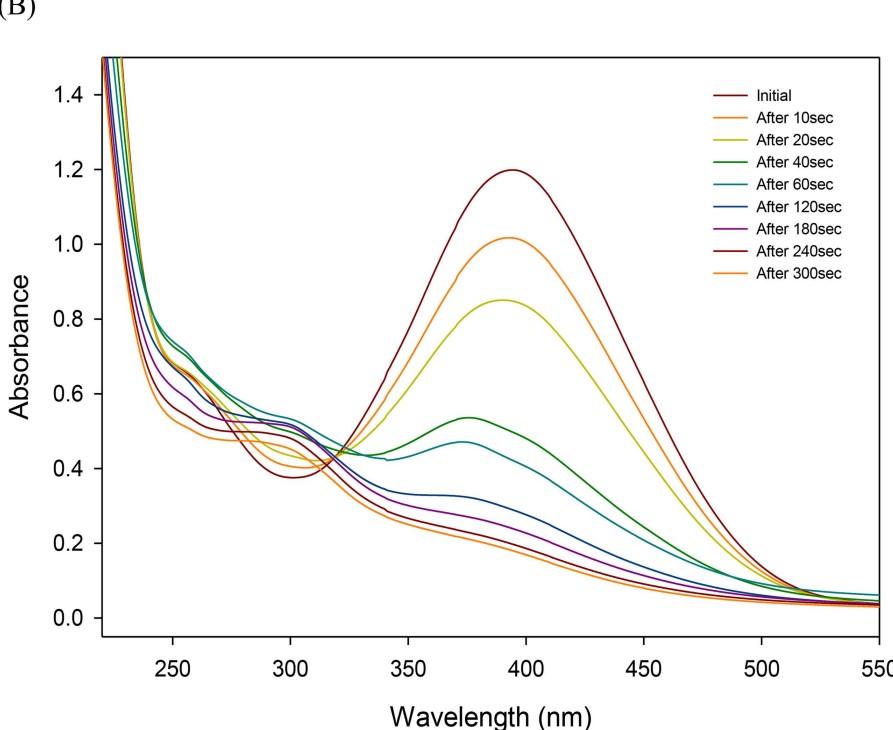

**Fig 3. Bleaching of RNO with yeast under UV irradiation (A) in 80 ppm HClO and (B) in 150 ppm HClO.**

**Table 1. Reduction of air suspended *Salmonella typhimurium* by individual treatment.**

| Treatment | | Single Conditions | |
|---|---|---|---|
| | | time/ppm | Reduction $\log_{10}$ CFU/m³ |
| Non yeast | UV only | 60sec | 4.06±0.15 |
| | HClO only | 80 | 4.24±0.14 |
| | | 120 | 4.56±0.12 |
| | | 150 | 4.95±0.07 |
| 5% Yeast extraction solution | UV only | 60sec | 3.33±0.10 |
| | HClO only | 80 | 3.14±0.12 |
| | | 120 | 3.64±0.07 |
| | | 150 | 3.99±0.05 |

$\log_{10}$ CFU/m³, respectively. The decline in UV efficacy is likely due to the deposition of organic matter on bacterial surfaces, which interferes with UVC absorption required for effective sterilization [36,37].

The bactericidal effects of slightly acidic electrolyzed water (SAEW) on microorganisms such as *Bacillus cereus*, *Listeria monocytogenes*, *Escherichia coli* O157:H7, and *Salmonella enterica* were not affected by the presence of high concentrations of carbohydrates or lipid compounds [38]. However, proteins have been shown to influence biocidal activity, as they are readily oxidized by hypochlorous acid [39].

Hydroxyl radicals (•OH) are highly reactive species that function as powerful oxidizing agents, capable of rapidly reacting with surrounding compounds [40]. Due to their extreme reactivity, •OH radicals have an extremely short diffusion distance in biological systems, as they are almost immediately neutralized upon contact with oxidizable substrates. Their average lifetime varies depending on the chemical composition of the environment [41,42].

Hydroxyl radicals exert bactericidal effects on both bacteria and viruses by inducing chemical modifications to essential biomolecules, including DNA, proteins, and lipids [43]. In the RNO bleaching experiments conducted in this study, UV irradiation was found to accelerate the decomposition of hypochlorous acid, leading to the rapid generation of •OH radicals. This mechanism enhanced sterilization efficiency, even in the presence of organic matter, regardless of the HOCl concentration.

### 3.3 Synergistic bactericidal effect of OH radical induced by HOCl under UV irradiation

When UV irradiation was applied for 30 and 60 seconds in combination with hypochlorous acid spraying for 60 seconds, the bacterial reduction ranged from 3.30±0.11 to 3.31±0.13 $\log_{10}$ CFU/m³ at 80 ppm, and from 3.86±0.07 to 3.87±0.08 $\log_{10}$ CFU/m³ at 120 ppm (Table 2). These results indicate that at lower concentrations (80 and 120 ppm), the duration of UV exposure did not significantly affect the sterilization efficacy. However, at the higher concentration of 150 ppm, an

**Table 2. Reduction of air suspended *Salmonella typhimurium* under combined conditions of HClO and UV irradiation.**

| Treatment | | Combined Conditions | | |
|---|---|---|---|---|
| | | ppm | UV Irradiation Time | |
| | | | 30sec | 60sec |
| | | | Reduction $\log_{10}$ CFU/m³ | Reduction $\log_{10}$ CFU/m³ |
| 5% Yeast extraction solution | UV+ HClO | 80 | 3.30±0.11 | 3.31±0.13 |
| | | 120 | 3.86±0.07 | 3.87±0.08 |
| | | 150 | 4.13±0.06 | 4.31±0.07 |

increase in UV irradiation time resulted in a greater sterilization effect, with bacterial reductions increasing from $4.13 \pm 0.06$ to $4.31 \pm 0.07$ $\log_{10}$ CFU/m³.

The reactivity of hydroxyl radicals (•OH) can be significantly reduced by the presence of various airborne compounds, such as small organic solvents, antioxidants, and natural substances [38–40]. In our study, tests using HOCl or UV alone demonstrated that the presence of organic matter substantially diminished the oxidative and bactericidal effectiveness of •OH generated by UV-irradiated hypochlorous acid, a finding consistent with recent research on radical scavenging mechanisms. Therefore, under real-world conditions, the actual disinfection efficiency may be lower than expected due to such interference. However, when HOCl and UV were applied simultaneously, the disinfection efficiency was minimally affected by the presence of organic matter.

The synergistic effect of hypochlorous acid and UV light under conditions where organic matter interferes with disinfection is supported by previous studies on surface sterilization. When UV light or 30 ppm hypochlorous acid were applied individually in the presence of organic matter, they achieved only 0.85 and 2.17 log reductions but the combined treatment resulted in a significantly enhanced reduction of 3.02 logs, demonstrating the synergistic bactericidal effect of HOCl and UV light even under organic load conditions [44]. These findings are consistent with our results, where UV efficacy alone was markedly reduced due to the shielding effect of organic matter.

The study of antibacterial mechanism of combining SAEW with UV light against *Staphylococcus aureus* showed that this combination reduces intracellular ATP levels and disrupts the bacterial electron transport chain through the introduction of hydrogen chloride [45–48]. In addition, hydroxyl radicals (•OH) generated during the process interact with the bacterial cell membrane, producing intracellular peroxides. This leads to the accumulation of reactive oxygen species (ROS), ultimately resulting in bacterial inactivation.

## 4. Conclusion

Livestock farms are highly susceptible to outbreaks of pathogenic microorganisms, necessitating effective and reliable airborne disinfection strategies. However, the presence of organic matter in real-world environments often interferes with disinfection efficacy. In this study, the synergistic antibacterial effects of ultraviolet (UV) irradiation and hypochlorous acid (HOCl) were investigated as a promising approach to overcome such challenges.

The study confirmed the generation of hydroxyl radicals (•OH) through the oxidation of RNO, particularly when HOCl was exposed to UV light. The formation of radicals was found to be concentration-dependent and increased with longer UV exposure times. Although organic matter—represented by yeast extract—significantly suppressed radical activity and disinfection performance, the combination of UV and HOCl consistently maintained higher bactericidal efficacy compared to either treatment alone. Notably, at 150 ppm HOCl, increasing UV exposure from 30 to 60 seconds further enhanced bacterial reduction, highlighting the importance of optimizing both chemical concentration and light exposure time.

In addition, the study demonstrated that the oxidative power of •OH radicals was strongly inhibited by organic compounds, as seen in the reduced RNO bleaching and sterilization values in the presence of yeast extract. Nevertheless, the co-application of HOCl and UV irradiation exhibited improved performance under organic-rich conditions, confirming the synergistic interaction between the two agents. This study underscores the potential applicability of HOCl combined with UV light as an effective airborne disinfection strategy for livestock environments. The approach is especially valuable in conditions where organic matter is present, offering a feasible and scalable solution for bioaerosol control in real-world agricultural and veterinary settings.

## Author contributions

**Conceptualization:** Younghee Kim, Hwa Yong Lee.

**Data curation:** Younghee Kim, Hwa Yong Lee.

**Formal analysis:** Han Bit Lee.

**Funding acquisition:** Hwa Yong Lee.

**Investigation:** Han Bit Lee.

**Methodology:** Hwa Yong Lee, Han Bit Lee.

**Writing – original draft:** Younghee Kim.

**Writing – review & editing:** Younghee Kim.

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
