## [Decision Letter · Decision Letter 0]

9 Apr 2025

PONE-D-25-15698Synergistic antibacterial effect of hydroxyl radical produced by combined hypochlorous acid and UV.PLOS ONE

Dear Dr. Kim,

Thank you for submitting your manuscript to PLOS ONE. After careful consideration, we feel that it has merit but does not fully meet PLOS ONE’s publication criteria as it currently stands. Therefore, we invite you to submit a revised version of the manuscript that addresses the points raised during the review process.

Please submit your revised manuscript by May 24 2025 11:59PM. If you will need more time than this to complete your revisions, please reply to this message or contact the journal office at plosone@plos.org . Please include the following items when submitting your revised manuscript:

We look forward to receiving your revised manuscript.

Kind regards,

Semira Galijasevic

Academic Editor

PLOS ONE

“This work was supported by Korea Institute of Planning and Evaluation for Technology in Food, Agriculture and Forestry (IPET) through Technology Commercialization Support Program, funded by Ministry of Agriculture, Food and Rural Affairs (MAFRA) (821034-3).”

“Hwa yong Lee reports financial support was provided by Korea Institute of Planning and Evaluation for Technology in Food, Agriculture and Forestry”

Reviewers' comments:

Reviewer's Responses to Questions

**Comments to the Author**

1. Is the manuscript technically sound, and do the data support the conclusions?

Reviewer #1: Partly

Reviewer #2: Yes

2. Has the statistical analysis been performed appropriately and rigorously? 

Reviewer #1: N/A

Reviewer #2: I Don't Know

3. Have the authors made all data underlying the findings in their manuscript fully available?

Reviewer #1: Yes

Reviewer #2: Yes

4. Is the manuscript presented in an intelligible fashion and written in standard English?

Reviewer #1: Yes

Reviewer #2: Yes

5. Review Comments to the Author

Reviewer #1: The submitted manuscript is original and proposes solutions to problems that lack new technologies to solve them. However, the authors do not consider the impact that the application of their proposal based on UVC/HOCl processes could have on the formation of toxic chlorinated species that could be introduced into living beings, causing a greater problem. I think the authors should expand their study by addressing the byproducts that are being produced, especially because they highlight the effect of organic matter in the process and ignore the generation of chlorinated byproducts.

Additionally, the authors should modify the introduction to address more the state of the art of the oxidative radicals of the radiation/chlorine system in the inactivation of pathogens, if the novelty of the study lies in the role of hydroxyl radicals in the inactivation of bacteria.

Reviewer #2: 1.More findings of the work should be added to the abstract.

2.Please check and complete conditions and procedures in experimental section and all procedures should be written clearly in detail.

3.Please clarify the innovations of this work in the introduction section.

4.Include the purity of the used chemicals in the text.

5.Discuss the effect of scavenger to confirm the role of OH free radical and other reactive species.

6.The authors need to provide the standard antibiotics details in the antibacterial activity assay.

7.Deeply check and polish English language level, there are some spelling and grammatical errors.

6. PLOS authors have the option to publish the peer review history of their article (what does this mean? ). If published, this will include your full peer review and any attached files.

**Do you want your identity to be public for this peer review?** For information about this choice, including consent withdrawal, please see our Privacy Policy .

Reviewer #1: No

Reviewer #2: No

---

## [Author Response · Author response to Decision Letter 1]

24 Jul 2025

Dear reviewers

I appreciate the kind comments.

Here are the answers to those comments.

Reviewer #1:

The submitted manuscript is original and proposes solutions to problems that lack new technologies to solve them. However, the authors do not consider the impact that the application of their proposal based on UVC/HOCl processes could have on the formation of toxic chlorinated species that could be introduced into living beings, causing a greater problem. I think the authors should expand their study by addressing the byproducts that are being produced, especially because they highlight the effect of organic matter in the process and ignore the generation of chlorinated byproducts. Additionally, the authors should modify the introduction to address more the state of the art of the oxidative radicals of the radiation/chlorine system in the inactivation of pathogens, if the novelty of the study lies in the role of hydroxyl radicals in the inactivation of bacteria.

Answer : I agree with the effect of byproducts produced by UVC/HOCl process. The process produces OH radical which can affect living organisms and kill viruses and bacteria that we call as disinfection. The production and effect of chlorinated byproducts was mentioned in “Introduction” section and previous studies were cited for potential use of the process. In addition, the result was discussed with previous studies which showed adverse effect on living organisms.

Reviewer #2:

1. More findings of the work should be added to the abstract.

Answer : Abstract was revised to address more findings of this study.

2.Please check and complete conditions and procedures in the experimental section and all procedures should be written clearly in detail.

Answer : The experimental procedure and test conditions were clearly described in the experimental section.

3.Please clarify the innovations of this work in the introduction section.

Answer : The meaning of this study and aims were more clearly mentioned to clarify the innovation of the study.

4.Include the purity of the used chemicals in the text.

Answer : The purity of the chemicals used is included in the text.

5.Discuss the effect of scavenger to confirm the role of OH free radical and other reactive species.

Answer : To investigate the effect of radical consumption and the reduction in disinfection effect due to organic substances in the air, yeast was used as a scavenger for the radical effect. The phenomenon of a decrease in the disinfection effect of Salmonella bacteria when airborne yeast is present was observed, and this result was examined by comparing it with the scavenger role suggested in previous research papers and revised in result and discussion section.

6.The authors need to provide the standard antibiotics details in the antibacterial activity assay.

Answer : In this study the standard antibiotics as the positive control was not used. We investigated the hindrance of organic material

7.Deeply check and polish English language level, there are some spelling and grammatical errors.

Answer : English expressions are reviewed from the native speaker.

---

## [Decision Letter · Decision Letter 1]

23 Sep 2025

Synergistic antibacterial effect of hydroxyl radicals generated by the combination of hypochlorous acid and UV irradiation

PONE-D-25-15698R1

Dear Dr. Kim,

We’re pleased to inform you that your manuscript has been judged scientifically suitable for publication and will be formally accepted for publication once it meets all outstanding technical requirements.

Kind regards,

Abayeneh Girma

Academic Editor

PLOS ONE

Additional Editor Comments (optional):

Reviewer #1:

Reviewer #2:

Reviewer #3:

Reviewers' comments:

Reviewer's Responses to Questions

**Comments to the Author**

1. If the authors have adequately addressed your comments raised in a previous round of review and you feel that this manuscript is now acceptable for publication, you may indicate that here to bypass the “Comments to the Author” section, enter your conflict of interest statement in the “Confidential to Editor” section, and submit your "Accept" recommendation.

Reviewer #1: All comments have been addressed

Reviewer #2: All comments have been addressed

Reviewer #3: All comments have been addressed

2. Is the manuscript technically sound, and do the data support the conclusions?

Reviewer #1: No

Reviewer #2: Yes

Reviewer #3: Yes

3. Has the statistical analysis been performed appropriately and rigorously? 

Reviewer #1: No

Reviewer #2: Yes

Reviewer #3: Yes

4. Have the authors made all data underlying the findings in their manuscript fully available?

Reviewer #1: Yes

Reviewer #2: Yes

Reviewer #3: Yes

5. Is the manuscript presented in an intelligible fashion and written in standard English?

Reviewer #1: Yes

Reviewer #2: Yes

Reviewer #3: Yes

6. Review Comments to the Author

Reviewer #1: Although the proposed disinfection system is interesting, the authors cannot simplify a system like UV/chlorine to the sole action of hydroxyl radicals. This system has already been extensively studied, and the coexistence of Cl radicals and ClO radicals along with hydroxyl radicals is evident. Therefore, I do not recommend its publication.

Reviewer #2: I have reviewed the revised version of the manuscript . The authors have addressed all the reviewer comments satisfactorily, and I am pleased to confirm that the manuscript is now suitable for publication in its current form.

Reviewer #3: All the comments raised by the reviewer have been responded well by the authors in the revised manuscript with Few mistakes.

7. PLOS authors have the option to publish the peer review history of their article (what does this mean? ). If published, this will include your full peer review and any attached files.

**Do you want your identity to be public for this peer review?** For information about this choice, including consent withdrawal, please see our Privacy Policy .

Reviewer #1: No

Reviewer #2: No

Reviewer #3: No

---

## [Editor Report · Acceptance letter]

PONE-D-25-15698R1

PLOS ONE

Dear Dr. Kim,

I'm pleased to inform you that your manuscript has been deemed suitable for publication in PLOS ONE. Congratulations! Your manuscript is now being handed over to our production team.

Kind regards,

on behalf of

Dr. Abayeneh Girma

Academic Editor

PLOS ONE